# Possible Involvement of the Upregulation of ΔNp63 Expression Mediated by HER2-Activated Aryl Hydrocarbon Receptor in Mammosphere Maintenance

**DOI:** 10.3390/ijms232012095

**Published:** 2022-10-11

**Authors:** Yuichiro Kanno, Nao Saito, Naoya Yamashita, Kazuki Ota, Ryota Shizu, Takuomi Hosaka, Kiyomitsu Nemoto, Kouichi Yoshinari

**Affiliations:** 1Laboratory of Molecular Toxicology, School of Pharmaceutical Sciences, University of Shizuoka, 52-1, Yada, Suruga-ku, Shizuoka 422-8526, Japan; 2Department of Molecular Toxicology, Faculty of Pharmaceutical Sciences, Toho University, 2-2-1 Miyama, Funabashi, Chiba 274-8510, Japan

**Keywords:** aryl hydrocarbon receptor, cancer stem cells, breast cancer, ΔNp63

## Abstract

Cancer stem cells (CSCs) contribute to the drug resistance, recurrence, and metastasis of breast cancers. Recently, we demonstrated that HER2 overexpression increases mammosphere formation via the activation of aryl hydrocarbon receptor (AHR). In this study, the objective was to identify the mechanism underlying mammosphere maintenance mediated by HER2 signaling-activated AHR. We compared the chromatin structure of AHR-knockout (AHRKO) HER2-overexpressing MCF-7 (HER2-5) cells with that of wild-type HER2-5 cells; subsequently, we identified *TP63*, a stemness factor, as a potential target gene of AHR. ΔNp63 mRNA and protein levels were higher in HER2-5 cells than in HER2-5/AHRKO cells. Activation of HER2/HER3 signaling by heregulin treatment increased ΔNp63 mRNA levels, and its induction was decreased by AHR knockdown in HER2-5 cells. The results of the chromatin immunoprecipitation assay revealed an interaction between AHR and the intronic region of *TP63*, which encodes ΔNp63. A luciferase reporter gene assay with the intronic region of *TP63* showed that AHR expression increased reporter activity. Collectively, our findings suggest that HER2-activated AHR upregulates ΔNp63 expression and that this signaling cascade is involved in CSC maintenance in HER2-expressing breast cancers.

## 1. Introduction

Breast cancer (BC) is the most commonly diagnosed neoplasm and the leading cause of cancer-related death in women worldwide [1,2]. The cancer stem cell (CSC) theory provides new insights into cancer therapies [3,4]. CSCs account for a relatively small population of tumor cells that are mainly identified by their properties of self-renewal, slow cell division, and tumor- and metastasis-initiating capacity [5,6]. Consequently, CSCs contribute to the drug resistance, recurrence, and metastasis of BCs. CSCs are commonly identified as a CD44^high^/CD24^low^ or aldehyde dehydrogenase-high population or a population with tumorsphere formation capacity [7]. Although the mechanisms underlying the self-renewal and tumor-initiation capacities of breast CSCs remain unclear, CSCs have been suggested as potential targets for radical cancer treatment.

HER2 belongs to the HER family of receptor tyrosine kinases, including HER1, HER3, and HER4 [8]. Upon ligand–HER binding via the extracellular ligand-binding domain of HER, the receptor undergoes homodimerization or heterodimerization, which activates its downstream signaling pathways, such as the mitogen-activated protein kinase pathway and phosphoinositide-3-kinase/Akt signaling pathway [8]. Although ligands for HER2 are not known, HER2 activation via heterodimerization with other HER receptors and/or mutations leading to aggressive tumor growth and poor clinical outcomes have been reported [9].

Cells with CSC-like properties have higher HER2 levels, which are controlled by Notch1 signaling, than bulk cells [10]. HER2 overexpression is associated with the presence of breast CSCs. For example, HER2 expression was found to be increased in CD44^high^/CD24^low^ luminal MCF-7 BC cells that express ALDH. The treatment of luminal BC cells with the humanized anti-HER2 monoclonal antibody trastuzumab decreases the population of CD44^high^/CD24^low^ cells or those highly expressing ALDH, as well as their mammosphere-formation ability [11,12]. HER2 overexpression in BC cell lines increases CSC populations [13]. These results suggest that HER2 is a positive factor for breast CSC self-renewal.

Aryl hydrocarbon receptor (AHR) is a ligand-activated transcription factor that is a member of the basic helix-loop-helix/Per-ARNT-Sim family [14]. Numerous studies have demonstrated that dioxin and dioxin-like chemicals, including 2,3,7,8-tetrachlorodibenzo-p-dioxin and polycyclic aromatic hydrocarbons, function as AHR agonists [15]. After ligand binding, AHR is translocated from the cytoplasm into the nucleus, where it forms a heterodimer with the AHR nuclear translocator (ARNT) to bind to AHR-response elements (AHREs). AHR regulates the expression of various genes, including *CYP1A1*, *CYP1A2*, and *CYP1B1* [16].

We previously reported that HER2 activation increases AHR expression and transcriptional activity in an agonist-independent manner in MCF-7 cells [17,18]. Furthermore, HER2 signaling-activated AHR increases the expression of inflammatory cytokines, such as IL-6 and IL-8, and mammosphere formation in these cells [17,18]. These results suggest that AHR regulates CSC functions in BCs.

The p63 isoform ΔNp63 was found to be a master regulator of HER2-subtype breast CSCs in a tumorigenesis model of transgenic mice with mouse mammary tumor virus (MMTV)-driven ERBB2 (active homolog of human HER2) [19,20]. ΔNp63 is encoded by the *TP63* gene, which produces various isoforms using different transcription start sites, such as the transactivating (TA) isoform and the N-terminally truncated (ΔN) isoforms, including ΔNp63. The knockdown of TP63 in cells of MMTV-ERBB2 tumor-derived mammospheres, which dominantly express the ΔNp63 isoform but not the TA isoform, decreases mammosphere formation in vitro and in tumor formation in vivo [20]. In addition, ectopic expression of ΔNp63 in MCF-7 cells increases the CSC population and mammosphere formation [21]. 

Understanding the self-renewal mechanism of CSCs is crucial to identifying new therapeutic targets for BC treatment. In this study, we investigated differences in the open chromatin regions between AHR-knockout and parental MCF-7 cells that overexpress HER2 to understand the mechanism by which AHR regulates the expression of genes related to the self-renewal of CSCs in HER2-overexpressing BCs. Here, we report the importance of the HER2-AHR-ΔNp63 axis in the self-renewal of CSCs in HER2-overexpressing BCs.

## 2. Results

### 2.1. Knockout of AHR Downregulates Mammosphere Formation and ΔNp63 Expression in HER2-Overexpressing Breast Cancer Cells

First, to investigate the role of AHR in the function of CSCs from HER2-overexpressing breast cancer cells, we compared the mammosphere formation efficiency of HER2-5 cells with that of HER2-5/AHRKO cells. HER2-5 is a previously established HER2-expressing MCF-7-derived stable cell line, and HER2-5/AHRKO is a HER2-5-derived cell line in which AHR has been knocked out using the CRISPR/Cas9 technique [17,18]. As expected, AHR was not observed in HER2-5/AHRKO cells (Figure 1A). AHR is highly expressed and constitutively activated in HER2-5 cells compared to levels in MCF-7 cells [18]. We previously showed that the mammosphere formation efficiency of HER2-5 cells is better than that of MCF-7 cells. As shown in Figure 1B, the mammosphere formation efficiency of HER2-5/AHRKO cells was worse than that of HER2-5 cells. These observations indicate that AHR is involved in mammosphere formation in HER2-overexpressing BC cells, as previously reported [18]. Thus, we used HER2-5 cells as a model to address the mechanism through which HER2 signaling-activated AHR contributes to the maintenance of mammospheres. 

Next, we investigated whether AHR engages in the chromatin relaxation of gene enhancers associated with the self-renewal of HER2-5 cells. We compared chromatin structures in HER2-5 and HER2-5/AHRKO cells by performing an assay for transposase-accessible chromatin using sequencing (ATAC-seq). ATAC-seq revealed 80,077 and 57,612 enriched peaks (i.e., open chromatin regions) in HER2-5 and HER2-5/AHRKO cells, respectively, and the chromatin positions of 50,265 peaks for HER2-5 cells matched those in HER2-5/AHRKO cells (Figure 1C); thus, unmatched peaks were changed by knocking out AHR in HER2-5 cells. These results indicated that HER2-5 cells had more regions of increased chromatin accessibility than HER2-5/AHRKO cells, suggesting that AHR engages in chromatin relaxation in HER2-5 cells. 

The peak sizes of the matching regions between these cells were then compared using the DESeq2 algorithm. The numbers of regions with increased (“upregulated”) and decreased (“downregulated”) peak areas after AHR knockout were 1519 and 8593, respectively. We annotated the genes close to the differentially regulated peaks using DA-VID annotation databases with criteria of 1.5-fold for upregulation and 0.5-fold for downregulation (Figure 1D). The term “pathways in cancer” was most frequently enriched for downregulated chromatin regions. Pathways related to xenobiotic metabolism, which is a classical function of AHR, were enriched with both upregulated and downregulated peaks.

ΔNp63 is known as a master regulator of HER2-subtype breast CSCs [19,20]. Since “downregulated” peak areas corresponding to the *TP63* gene were observed based on the results shown in Figure 1C, we targeted the ΔNp63-encoding gene in this study. First, we compared the ΔNp63 expression level in HER2-5 cells with that in HER2-5/AHRKO cells. As expected, ΔNp63 mRNA levels were higher in HER2-5 cells than in HER2-5/AHRKO cells (Figure 2A). Moreover, ΔNp63 protein levels were higher in HER2-5 cells than in HER2-5/AHRKO cells (Figure 2B). These results suggest that AHR upregulates ΔNp63 expression in HER2-overexpressing BC cells.

### 2.2. HRG/HER2 Signaling Induces ΔNp63 Expression via AHR Activation

The HER2/HER3 heterodimer plays a major role in promoting the proliferation and migration of HER2-positive BCs [23]. In addition, we previously reported that HER2/HER3 signaling, activated by treatment with the HER3 ligand heregulin-β1 (HRG), promotes AHR expression and nuclear translocation in HER2-5 cells [17,18]. Thus, we investigated whether HER2/HER3 signaling increases ΔNp63 mRNA levels via AHR in HER2-5 cells. 

HER2-5 cells were cultured in low-serum medium (2% charcoal-stripped fetal bovine serum, 2% csFBS). Under this condition, phosphorylation of the HER2/HER3 downstream proteins ERK1/2 and Akt was not detected by Western blotting; however, phosphorylation was detected after stimulation with HRG (50 ng/mL) for 15 min (Appendix A). These observations suggest that HER2 activation does not occur under the low-serum conditions. Next, the cells cultured in low-serum medium were treated with HRG for 3, 6, 12, or 24 h. ΔNp63 mRNA levels increased following HRG treatment, with the highest level at 3 h (Figure 3A). Furthermore, ΔNp63 protein levels also increased following HRG treatment (Figure 3B). Subsequently, we knocked down AHR using siRNA to investigate the role of AHR in the HRG-induced increase in ΔNp63 mRNA levels. As shown in Figure 3C, the HRG-mediated increase in ΔNp63 mRNA levels was suppressed by AHR knockdown. Similarly, the HRG-mediated increase in ΔNp63 mRNA levels was suppressed by AHR antagonist treatment (Appendix A). These results suggest that AHR participates in the HER2/HER3-mediated expression of ΔNp63. However, the HRG-mediated increase in ΔNp63 mRNA levels was not completely suppressed in HER2-5/AHRKO cells (Appendix A). Previously, we reported that the signaling activation of AHR via HER2 is mediated by the MEK cascade. To identify the mechanism underlying ΔNp63 mRNA expression induced by HER2 signaling, HER2-5 cells were pretreated with specific inhibitors of MEK and AKT, namely PD0325901 and MK-2206, respectively (Appendix A). Following HRG treatment, HRG-induced mRNA expression of ΔNp63 was completely inhibited by pre-treatment with PD0325901, but not MK-2206 (Figure 3D). These results suggest that ΔNp63 induction via HER2 is regulated by AHR and other molecules through MEK.

### 2.3. HRG-Activated AHR Binds to the Enhancer Region of TP63

We next investigated the mechanism by which AHR increased ΔNp63 expression. Figure 4A shows the tracks of ATAC-seq analyses of HER2-5 and HER2-5/AHRKO cells. *TP63* encodes two major isoforms of p63, TAp63 and ΔNp63. ΔNp63 regulates CSC self-renewal in HER2-overexpressing BC cells [19,20]. Therefore, we investigated whether AHR regulates ΔNp63 expression. First, to investigate the recruitment of AHR to the promoter region of *TP63*, we performed chromatin immunoprecipitation (ChIP) analysis for regions R1, R2, R3, and R4 (Figure 4A), which contained differentially enriched peaks between HER2-5 and HER2-5/AHRKO cells based on the ATAC-seq analysis. R1 and R3 are regions distal and proximal to the transcription start site of TAp63- and ΔNp63-encoding mRNAs, respectively. R2 and R4 are the intronic regions of *TP63*. 

The ChIP assay demonstrated that AHR was recruited to R4 in HER2-5 cells but not in negative control HER2-5/AHRKO cells. Additionally, AHR was not recruited to R1, R2, or R3 in either cell line (Figure 4B). As expected, AHR was recruited to the *CYP1B1* enhancer region only in HER2-5 cells (Figure 4B). 

Subsequently, we investigated whether AHR recruitment to R4 is induced by HER2 signaling in HER2-5 cells. Under low serum conditions, AHR did not bind to the R4 region as well as it did to the R3 region; however, HRG treatment for 3 h increased AHR recruitment to R4 but not to R3 (Figure 4C). These results suggest that HER2 signaling promotes the recruitment of AHR to the R4 region of *TP63*.

Finally, we investigated whether AHR acts as a transcriptional activator by binding to the R4 region of *TP63*. We constructed a luciferase reporter plasmid containing the R4 region, pGL4.24-TP63 R4 (Figure 4D). Our analysis using the JASPAR database [24] identified three putative AHREs in the R4 region; therefore, we also prepared a reporter plasmid, pGL4.24-TP63 R4 AHREm, with mutated AHREs (Figure 4D).

HER2-5/AHRKO cells were transfected with pGL4.24-TP63 R4 and AHR and/or ARNT expression plasmids, and luciferase reporter activity was measured. As shown in Figure 4E, reporter activity was increased by the expression of AHR and ARNT. This increase was suppressed by treatment with the AHR antagonist StemRegenin 1 (Figure 4F). Moreover, the increase in the luciferase reporter activity due to AHR and ARNT overexpression was not observed in the AHRE-mutated plasmid (Figure 4G). These results suggest that the AHREs in the R4 enhancer region are responsible for AHR-mediated ΔNp63 expression in HER2-5 cells.

## 3. Discussion

AHR overexpression is observed in BC, and increased nuclear localization of AHR is positively correlated with poor prognosis [25,26,27,28]. AHR is reportedly overexpressed in BC cells relative to expression in normal breast tissue and has been negatively associated with the histological type and p53 protein expression levels [29]. Furthermore, increasing evidence has shown that the repression of AHR inhibits the proliferation, invasion, and migration of BC cells [30,31]. These reports suggest that constitutively activated AHR contributes to cancer progression. We previously reported that HER2 overexpression, which is associated with a high degree of malignancy and poor prognosis for BC, enhances the CSC properties of BC cells via AHR [17,18].

An analysis of the role of AHR in HER2-overexpressing BC cells by performing ATAC-seq using HER2-5 and HER2-5/AHRKO cells suggested that HER2-activated AHR upregulates the expression of genes annotated as “pathways in cancers” (Figure 1D). AHR activation modulates the invasive properties of cancer cells. In melanoma cells, the activation of AHR promotes TNFα-dependent inflammation and metastasis [32]. Moreover, AHR activation by benzo[a]pyrene promotes cell migration, invasion, and epithelial–mesenchymal transition by upregulating the expression of long non-coding RNA in lung cancer [33].

In this study, we demonstrated that AHR binds to the R4 enhancer region of *TP63* in HER2-5 cells (Figure 4B), in which AHR was constitutively activated and accumulated in the nucleus [17,18]. Moreover, we showed that HRG-activated AHR was recruited to the R4 region (Figure 4C). Thus, growth factors such as HRG might activate AHR. Previously, we showed that mitogen-activated protein kinase/extracellular signal-regulated kinase kinase signaling enhances AHR expression and induces the accumulation of AHR in the nucleus [17]. These results suggest that HER2 signal-activated AHR binds to the R4 enhancer region of *TP63* in HER2-5 cells.

Estrogen receptor α (ERα) is a critical transcriptional regulator in BC cells. ΔNp63 mRNA transcription is regulated by various promoter regions. ERα binds to the estrogen response element located between −2,858 and −2,839 bp at the translation start site within the ΔNp63 promoter and induces the transcription of ΔNp63 mRNA [34]. ERα is known to exhibit crosstalk with AHR via various mechanisms [35,36,37]. For example, ligand-activated AHR forms a complex with ERα on the estrogen response element and potentiates the transactivation function of 17β-estradiol-unbound ERα. However, AHR represses 17β-estradiol-bound ERα [38]. Furthermore, AHR has ubiquitin ligase activity and pro-motes ERα degradation [36]. These facts suggest the involvement of ERα in the AHR-dependent expression of ΔNp63 via the R4 region. However, ERα was not found to bind to the R4 region based on in silico analysis using JASPAR (data not shown). In addition, mutations in the AHREs of the R4 region led to reduced AHR-dependent reporter gene expression (Figure 4G). Therefore, these results suggest that HRG-mediated ΔNp63 expression is independent of AHR–ERα crosstalk and is mediated by the direct binding of AHR to the R4 region.

ΔNp63 maintains the self-renewal capacity of mammary stem cells and the stemness of BC cells [19,39,40]. ΔNp63 enhances stemness through various stemness signaling pathways such as WNT, Hedgehog, and NOTCH. In addition, ΔNp63 directly controls the expression of stemness factors such as FZD7, SHH, GLI2, PTCH1, and NOTCH1 [20,21,41]. Based on these roles for ΔNp63, the upregulation of its expression via AHR could enhance CSC proliferation in BC cells. HER2 signaling can activate AHR and lead to constitutive ΔNp63 expression in HER2-overexpressing BC cells. This suggests that AHR controls mammosphere formation via ΔNp63 in HER2-overexpressing BC cells. As described previously herein, AHR expression might correlate with the poor prognosis and malignancy of BC. Because this study using HER2-overexpressing BC cells is limited, further study is required to determine whether AHR can increase ΔNp63 expression in BC cells of other subtypes.

AHR activation mediated by the endogenous ligand FICZ induces the expression of genes associated with migration, invasion, and stemness in triple-negative BC cells [42]. Furthermore, functional studies of AHR in CSCs have shown that it contributes to the chemoresistance of CSCs [43,44,45]. The expression of chemotherapy resistance-related transporter (such as ABCG2)- and enzyme (such as aldo-keto reductase 1C3 and ALDH1A1)-encoding genes, which are AHR-target genes, is increased in AHR-overexpressing CSCs [43,44,45]. Previous reports, including our studies, have established the anti-tumorigenic ability of agonist-activated AHR [46]. Agonist-mediated AHR activation represses mammosphere formation and reduces the ALDH-high population, suggesting the repressive effects of AHR on breast CSCs [47,48,49,50]. Moreover, we have previously reported the suppressive effect of AHR (depending on the agonist used) on mammosphere formation in BC cells [22,51]. These results suggest that AHR has both pro- and anti-tumorigenic activities in cancers and that an agonist can determine its activity. Thus, understanding how the ligand selectivity of AHR contributes to these pro- and anti-tumorigenic functions might provide information on AHR as a novel therapeutic target for cancer. In summary, the present results suggest that HER2 signaling-activated AHR maintains mammosphere formation by breast CSCs by inducing ΔNp63 expression. Therefore, AHR could be a potential therapeutic target for BC stem cell therapy.

## 4. Materials and Methods

### 4.1. Chemicals

StemRegenin 1 was purchased from Selleck Chemicals (Houston, TX, USA). Dimethyl sulfoxide was purchased from Wako Pure Chemical Industries (Osaka, Japan). HRG (EGF Domain) was purchased from Sigma-Aldrich (St. Louis, MO, USA) and dissolved in water.

### 4.2. Cell Lines and Cell Culture

Human BC MCF-7 cells were obtained from the Cell Resource Center for Biomedical Research (Institute of Development, Aging and Cancer, Tohoku University, Sendai, Japan) and cultured in Dulbecco’s modified Eagle’s medium (Wako Pure Chemical Industries) containing 10% fetal bovine serum (Sigma-Aldrich) and Antibiotic–Antimycotic (Thermo Fisher Scientific, Waltham, MA, USA) in a humidified atmosphere with 5% CO_2_ at 37 °C. HER2-overexpressing MCF-7 clone 5 (HER2-5) cells stably expressing HER2 have been previously established [18]. HER2-5 AHR-knockout cells (HER2-5/AHRKO) were generated as previously described [17]. For cell stimulation, HER2-5 cells were cultured in low serum medium containing 2% csFBS (Sigma-Aldrich) for 24 h and treated with recombinant HRG (50 ng/mL).

For the knockdown experiment, HER2-5 cells were transfected with AHR siRNA (Stealth™ RNAi, AHR-HSS 100338; Thermo Fisher Scientific) or control siRNA (Stealth™ RNAi siRNA Negative Control, Thermo Fisher Scientific) using Lipofectamine™ RNAiMAX transfection reagent (Thermo Fisher Scientific), according to the manufacturer’s instructions. After 24 h, the culture medium was changed to low-serum medium, and the cells were cultured for 21 h. Cells were stimulated with HRG (50 ng/mL) or vehicle (water) for 3 h.

### 4.3. Mammosphere Formation Assay

The mammosphere formation assay was performed as previously described [22]. Cells were seeded into 96-well ultra-low adherent plates at 1000 cells/well in MammoCult™ Medium (StemCell Technologies, Vancouver, Canada) and cultured for 7 days. The mammospheres were observed using the EVOS^®^ FL Cell Imaging System (Invitrogen, Thermo Fisher Scientific). Mammosphere formation efficiency was calculated as the number of spheres divided by the original number of seeded cells.

### 4.4. ATAC-Seq

ATAC-seq using HER2-5 and HER2-5/AHRKO cells was performed at Active Motif, Carlsbad, CA, USA. Only reads that were mapped to the human genome as matched pairs were used for analysis. Data from the HER2–5 and HER2-5/AHRKO cell samples were normalized to the same number of unique alignments (by downsampling to 28 million). Peaks were identified using MACS2 with a p-value cutoff of 1.0 × 10^−7^. For each possible pair-wise comparison, the shrunken log_2_ fold-change was calculated using the DESeq2 algorithm.

### 4.5. Reverse Transcription-Quantitative Polymerase Chain Reaction (RT-qPCR)

Total RNA from HER2-5 and HER2-5/AHRKO cells was isolated using ISOGEN II reagent (Nippon Gene, Tokyo, Japan), and cDNA was synthesized using the ReverTra Ace™ qPCR RT kit (Toyobo, Osaka, Japan) [18]. qPCR was performed using the GoTaq^®®^ qPCR Master Mix (Promega, Madison, WI, USA) and primers specific to ΔNp63 (forward, 5′-CTGGAAAACAATGCCCAGAC-3′ and reverse, 5′-GGGTGATGGAGAGAGAGCAT-3′) and beta-2-microglobulin (*B2M*; forward, 5′-TGCTGTCTCCATGTTTGATGT-3′ and reverse, 5′-TCTCTGCTCCCCACCTCTAAG-3′). The ΔNp63 mRNA levels were normalized to those of *B2M*.

### 4.6. Western Blotting

Cells were harvested and lysed in sample buffer (125 mM Tris-HCl, 4% sodium dodecyl sulfate, 20% glycerol, 1 mM dithiothreitol, and 0.01% bromophenol blue; pH 6.8). The protein concentration in the whole-cell lysates was determined using the Pierce™ 660 nm protein assay reagent (Thermo Fisher Scientific) and the Ionic Detergent Compatibility Reagent for Pierce™ 660 nm Protein Assay Reagent (Thermo Fisher Scientific). Equal amounts of whole-cell lysates were resolved by sodium dodecyl sulfate-polyacrylamide gel electrophoresis. Subsequently, the resolved proteins were transferred onto a polyvinylidene fluoride membrane (Immobilon^®^-P; Merck Millipore, Billerica, MA, USA) and probed with primary antibodies against AHR (D5S6H; Cell Signaling Technology, Danvers, MA, USA, 1:2000), ΔNp63 (E6Q3O; Cell Signaling Technology, 1:2000), or α-tubulin (Medical & Biological Laboratories, Nagoya, Japan, 1:20,000), followed by horseradish peroxidase-conjugated secondary antibodies against immunoglobulin G (IgG, Cell Signaling Technology, 1:10,000). The membrane was incubated with the Immobilon^®^ Crescendo Western HRP substrate (Merck Millipore, Billerica, MA, USA). Immunoreactive bands were visualized using the WSE-6100 LuminoGraph I (Atto, Tokyo, Japan) and analyzed using ImageJ software (version 1.52a, National Institutes of Health, Bethesda, MD, USA; https://imagej.nih.gov/ij/, accessed on 31 July 2022).

### 4.7. ChIP Assay

HER2-5 and HER2-5/AHRKO cells were seeded into 100 mm culture dishes and incubated for 48 h. Subsequently, a ChIP assay was performed using the SimpleChIP^®^ Plus Sonication Chromatin IP kit (Cell Signaling Technology) according to the manufacturer’s protocol. Chromatin was digested using the Bioruptor (Cosmo Bio, Tokyo, Japan) and immunoprecipitated using anti-AHR antibodies (D5S6H; Cell Signaling Technology) or IgG (Cell Signaling Technology) as a control overnight at 4 °C. The precipitated DNA fragments were quantified by qPCR using the GoTaq^®^ qPCR Master Mix (Promega) and primers specific to ΔNp63 enhancer R1 (chr3:189573177+189573353; forward, 5′-CTGATGCAGTCCCATTTCCT-3′ and reverse, 5′-AGTGCTCCAGGCAAAAAGAA-3′), ΔNp63 enhancer R2 (chr3:189573177+189573353; forward, 5′-AAAAACATGGCTGCACTT′CC-3′ and reverse, 5′-GGTTCTCAGAGCCATGTGGT-3′), ΔNp63 enhancer R3 (chr3:189722302+189722464; forward, 5′-TTTGGAATTGGCAGGTAAGG-3′ and reverse, 5′-GAGCTCTAAGGCCCTCTGGT-3′), ΔNp63 enhancer R4 (chr3:189841878+189842185; forward, 5′-ACTGCGTGAGGCTTTGTCTTGGACC-3′ and reverse, 5′-GCAGTAGCAGCATGTCTGTCTCAGC-3′), and the *CYP1B1* enhancer (chr2:38076942-38077091; forward, 5′-TGTCAGGTGCCGTGAGAA-3′ and reverse, 5′-CGAACTTTATCGGGTTGAA-3′).

### 4.8. Plasmid Construction

Double-stranded DNA fragments of the R4 enhancer (+111995 from ΔNp63 transcription start site to +112566 of *TP63*) and the R4 enhancer with mutations at three AHREs (GCGTG to GATTG) in the human *TP63* gene were synthesized by Eurofins Genomics (Ebersberg, Germany). The 5′- and 3′-terminal ends of these synthesized fragments contained KpnI and XhoI restriction sites, respectively. KpnI- and XhoI-digested fragments were inserted into the pGL4.24[luc2P/minP] vector (Promega). The pCMV-3Tag-6 plasmid containing the AHR-coding sequence (pCMV-3Tag-6 AHR) has been reported previously [22]. The pCMV-3Tag-6 ARNT plasmid was constructed by inserting amplified human ARNT cDNA into pCMV-3Tag-6 (Stratagene, Los Angeles, CA, USA) at the EcoRV/XhoI restriction sites. Full-length human *ARNT* cDNA was amplified by PCR using specific primers (forward, 5′-ATCACCATGGCGGCGACTACTGCCAA-3′ and reverse, 5′-CGCCTCGAGCTATTCTGAAAAGGGGGGAA-3′) and a cDNA solution prepared from MCF-7 cells.

### 4.9. Luciferase Reporter Gene Assay

HER2-5/AHRKO cells were transfected with the reporter and expression plasmids and the *Renilla* luciferase pGL4.74[hRluc/TK] plasmid (Promega; used as an internal standard) using the reverse transfection method with PEI Max^®^ (Polysciences, Warrington, PA, USA). After transfected HER2-5/AHRKO cells were incubated overnight at 37 °C, the cells were treated with the AHR antagonist StemRegenin 1 (10 μM) for 24 h. Subsequently, the cells were harvested, and luciferase activity was measured using the Dual-Luciferase^®^ Reporter Assay System (Promega). The activity of firefly luciferase was normalized to that of *Renilla* luciferase.

### 4.10. Statistical Analysis

Data of two groups were compared using a Student’s *t* test, whereas those of three or more groups were compared using one-way analysis of variance followed by Dunnett’s or Tukey’s test. All statistical analyses were conducted using KaleidaGraph (version 4.1.1; Synergy Software, Eden Prairie, MN, USA); *p*-values < 0.05 were considered statistically significant.

## Figures and Tables

**Figure 1 ijms-23-12095-f001:**
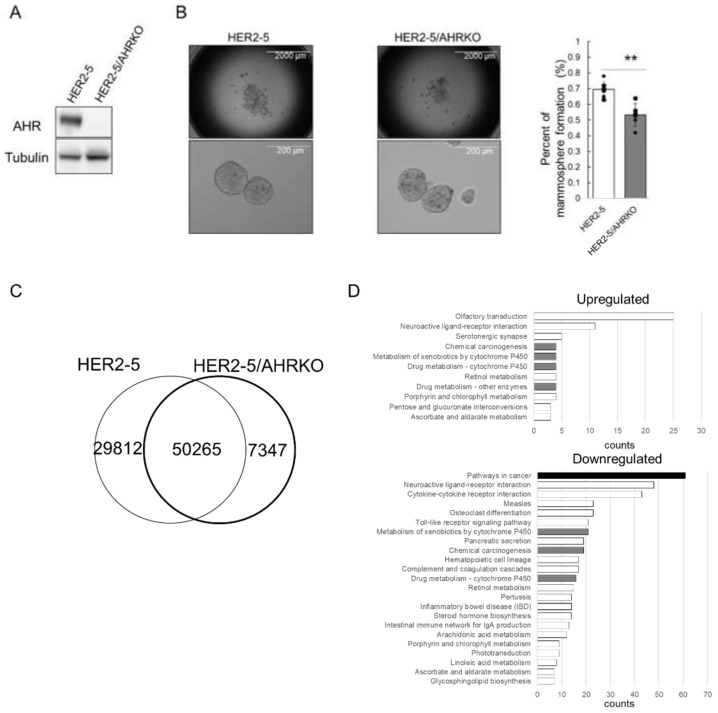
Difference in mammosphere formation efficiency and accessible chromatin regions between HER2-5 and HER2-5/AHRKO cells. (**A**) Whole-cell lysates (10 µg) were resolved by sodium dodecyl sulfate-polyacrylamide gel electrophoresis, and proteins were detected by immunoblot analysis using antibodies against AHR and α-tubulin. (**B**) HER2-5 cells and HER2-5/AHRKO cells were cultured under non-adherent conditions for 7 days. The mammospheres were observed under a light microscope with 2× magnification (scale bar is 2000 μm) and 20× magnification (scale bar is 200 μm). The percentage of cells contributing to mammosphere formation was calculated as the number of spheres divided by the original number of seeded cells [22]. Each column represents the mean of six wells. The depicted results are representative of two independent biological experiments. ** *p* < 0.01 (Student’s *t* test). (**C**) ATAC-seq was performed using HER2-5 cells and HER2-5/AHRKO cells. The overlap peaks are presented in a VENN diagram. The diagram shows the number of merged regions that contain peaks from the samples in each of the overlapping categories. (**D**) Upregulated and downregulated GO terms for biological processes enriched among differentially expressed genes were annotated using the DAVID tool.

**Figure 2 ijms-23-12095-f002:**
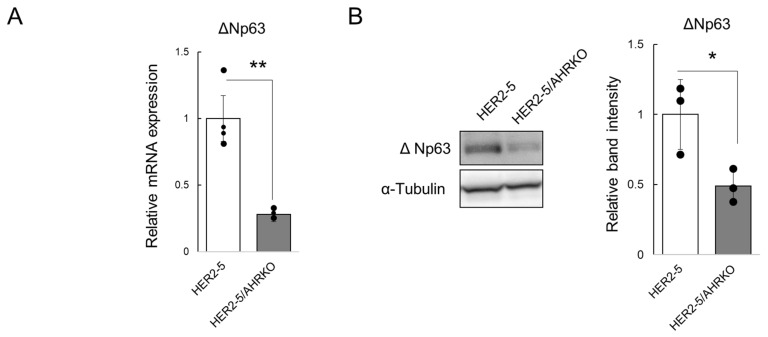
Downregulation of ΔNp63 mRNA and protein levels in HER2-5/AHRKO cells. (**A**) ΔNp63 mRNA levels were measured by RT-qPCR analysis. ΔNp63 mRNA levels were normalized to those of *B2M*. Values are expressed as relative values (those in HER2-5 cells were set at 1) and the mean ± S.D. (*n* = 4). ** *p* < 0.01 (Student’s *t* test). Each dot represents individual data points of quadruplicate measurements. The depicted results are representative of two independent biological experiments. (**B**) Whole-cell lysates (10 µg) were resolved by sodium dodecyl sulfate-polyacrylamide gel electrophoresis, and proteins were detected by immunoblot analysis using antibodies against ΔNp63 and α-tubulin. The intensity of ΔNp63 and α-tubulin protein bands was measured using ImageJ software. ΔNp63 levels were normalized to those of α-tubulin and are expressed as relative values (those in HER2-5 cells were set at 1) based on the mean ± S.D. (*n* = 3). Each dot represents independent biological data points of triplicate measurements. * *p* < 0.05 (Student’s *t* test).

**Figure 3 ijms-23-12095-f003:**
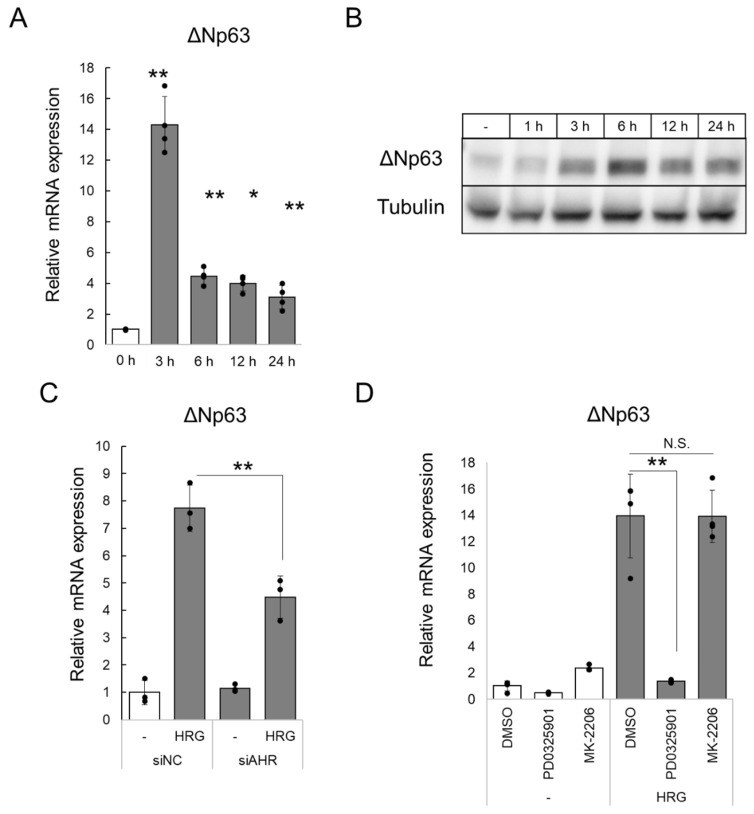
Upregulation of ΔNp63 expression via AHR mediated by HRG. (**A**) HER2-5 cells were cultured in medium with low serum (2% charcoal-stripped fetal bovine serum, csFBS DMEM) for 24 h and treated with HRG (50 ng/mL) for 3–24 h. Subsequently, the cells were harvested, and ΔNp63 mRNA levels were determined by RT-qPCR analysis and normalized to those of *B2M* mRNA. The relative mRNA levels are expressed as the average of the fold-induction relative to that in the control (0 h) (mean ± S.D., *n* = 4). ** *p* < 0.01; * *p* < 0.05 (Dunnett’s test vs. 0 h). The depicted results are representative of two independent biological experiments. (**B**) HER2-5 cells were cultured in medium with low serum (2% csFBS DMEM) for 24 h and treated with HRG (50 ng/mL) for 1–24 h. Subsequently, the cells were harvested, whole-cell lysates (10 µg) were resolved by sodium dodecyl sulfate-polyacrylamide gel electrophoresis, and proteins were detected by immunoblot analysis using antibodies against ΔNp63 and α-tubulin. (**C**) HER2-5 cells were transfected with siRNA targeting AHR (siAHR, 10 nM) or control siRNA (siNC, 10 nM). After 24 h, the culture medium containing 10% FBS was changed to medium containing low serum (2% csFBS), and the cells were incubated for 21 h. The cells were stimulated with HRG (50 ng/mL) or vehicle (water) for 3 h, and relative mRNA levels were determined (mean ± S.D., *n* = 3). ** *p* < 0.01 (Student’s *t* test). The results are representative of two independent biological experiments. (**D**) HER2-5 cells were cultured in medium with low serum (2% csFBS DMEM) for 24 h and pre-treated with PD0325901 (1 μM), MK-2206 (1 μM), or DMSO (solvent control) for 1 h. Then, the cells were stimulated with HRG (50 ng/mL) or vehicle (water) for 3 h, and relative mRNA levels were determined (mean ± S.D., *n* = 3). ** *p* < 0.01 (Student’s *t* test). The results are representative of two independent biological experiments.

**Figure 4 ijms-23-12095-f004:**
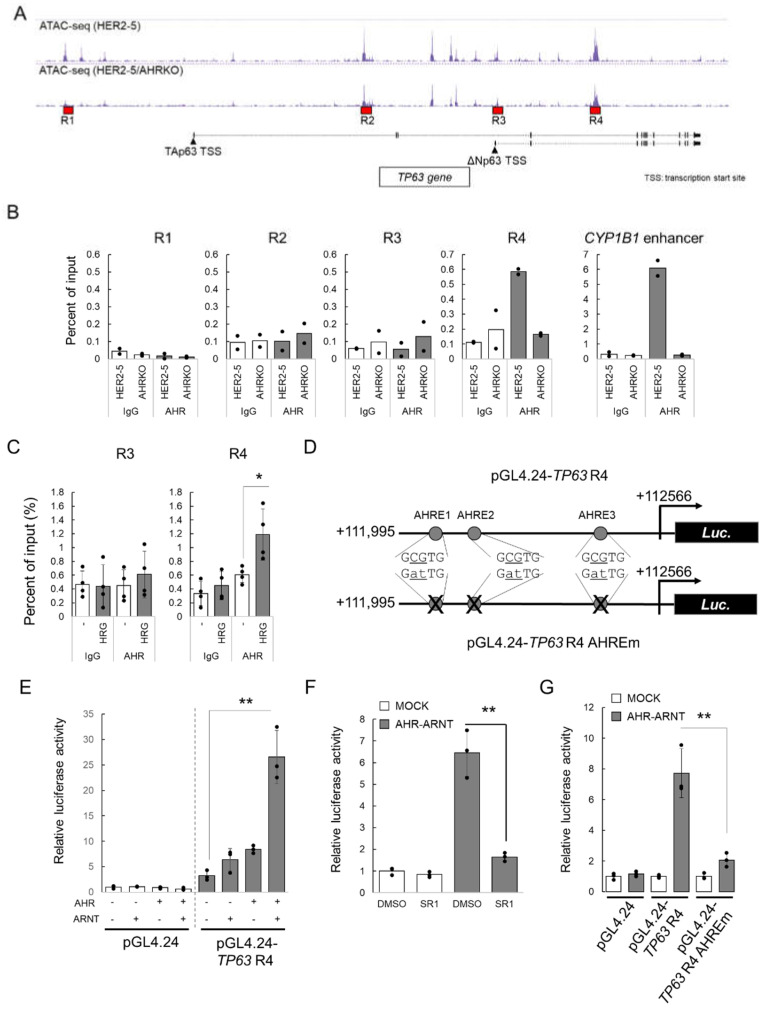
Binding of HRG-activated AHR to the enhancer region of *TP63*. (**A**) ATAC-seq tracks (HER2-5 and HER2-5/AHRKO cells) around *TP63* are shown. R1, R2, R3, and R4 regions were investigated by performing a chromatin immunoprecipitation (ChIP) assay. (**B**) HER2-5 and HER2-5/AHRKO cells were subjected to ChIP assays with anti-AHR and control IgGs. The AHR bound to DNA was quantified by qPCR with primers specific for R1, R2, R3, and R4, as well as the *CYP1B1* enhancer region. The bars represent the mean percentage of the input control of two independent experiments, and all data are expressed as closed cycles. (**C**) HER2-5 cells were cultured under low serum conditions (DMEM containing 2% charcoal-stripped fetal bovine serum, csFBS) for 24 h and treated with HRG (50 ng/mL) or vehicle (water) for 3 h. The cells were subjected to ChIP assays with anti-AHR and control IgGs. The amount of AHR bound to DNA was quantified by qPCR with primers specific for R3 and R4. The bars represent the mean percentage of the input control of four independent experiments and are expressed as the mean ± S.D. (*n* = 4). * *p* < 0.05 (Student’s *t* test). (**D**) Schematic illustration of pGL4.24-TP63 R4 and pGL4.24-TP63 R4 AHREm plasmids. Putative (circle, GCGTG) and mutated AHREs (GatTG) are indicated. (**E**) HER2-5/AHRKO cells were co-transfected with pGL4.24 or pGL4.24-TP63 R4 and pGL4.74[hRluc/TK] in combination with an expression plasmid for AHR and/or ARNT or an empty plasmid (MOCK), as indicated. The cells were harvested 48 h post-transfection, and luciferase activity was measured. The results are presented as relative luciferase activity (*n* = 3) relative to that in the pGL4.24/MOCK control cells (left open column), which was set to 1, and are expressed as the mean ± S.D. ** *p* < 0.01 (Tukey’s test). (**F**) HER2-5/AHRKO cells were co-transfected with pGL4.24-TP63 R4, pGL4.74[hRluc/TK], and expression plasmids for AHR and ARNT or an empty plasmid (MOCK), as indicated. After 24 h, the cells were treated with StemRegenin 1 (SR1, 10 µM) or 0.1% dimethyl sulfoxide (DMSO, solvent control) for 24 h. The cells were harvested, and the luciferase activity was measured. The results are presented as relative luciferase activity (*n* = 3) relative to that in the vehicle/MOCK control cells, which was set to 1, and are expressed as the mean ± S.D. ** *p* <0.01 (Student’s *t* test). (**G)** HER2-5/AHRKO cells were co-transfected with pGL4.24, pGL4.24-TP63 R4, or pGL4.24-TP63 R4 AHREm and pGL4.74[hRluc/TK], in combination with expression plasmids for AHR and ARNT or the empty plasmid (MOCK), as indicated. The cells were harvested, and the luciferase activity was measured. The results are presented as relative luciferase activity (*n* = 3) relative to that in cells transfected with MOCK (open column), which was set as 1, for each MOCK plasmid group and expressed as the mean ± S.D. ** *p* < 0.01 (Tukey’s test).

## Data Availability

Not applicable.

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
