# Peer review of "Possible Involvement of the Upregulation of ΔNp63 Expression Mediated by HER2-Activated Aryl Hydrocarbon Receptor in Mammosphere Maintenance"

_ijms, 2022, doi:10.3390/ijms232012095_

Round 1
Reviewer 1 Report
Authors may need to provide pathway diagram to assist reader's understanding.
Author Response
We have no response to reviewer 1.
Author Response
Response to reviewer 2
Overall, the manuscript is well structured. However, the manuscript needs a scientific proofread. Experimental data presentation is cohesive and consistent. However, the manuscript lacks adequate in vitro and in vivo functional evidence for the hypothesis. The authors should address the following comments to strengthen this manuscript.
1 Comment
1.1 From the abstract I learn that this manuscript demonstrated the HER2-AHR-ΔNp63 axis in CSC self-renewal and proliferation regulation. However, I cannot see any word in the title mentioning CSC. Herein the title is not good and summative for this manuscript.
Thank you for your suggestion. Accordingly, we have changed the title to “Possible involvement of the upregulation of ΔNp63 expression mediated by HER2-activated aryl hydrocarbon receptor in mammosphere maintenance”.
1.2 What is the basis of the nomenclature for HER2-5 cell?
We have added a description to define the nomenclature used for HER2-5 cells in the Materials and methods section.
Line 353: HER2-overexpressing MCF-7 clone 5 (HER2-5) cells stably expressing HER2 have been previously established.
1.3 Why did the authors choose HER2-overexpressing MCF7 (HER2-5) cells as cell models? Naturally, luminal B or HER2-overexpression subtype cells have high HER2 expression levels. Why not chose the natural cell models?
Thank you for your suggestion. We previously reported AHR overexpression and increasing mammosphere formation capacity after ectopically overexpressing HER2 in MCF-7 cells. Accordingly, we selected this as a good model. Moreover, we demonstrated that AHR knockdown decreases mammosphere formation in HER2-overexpressing MDA-MB 453 cells; we think the same result would be observed in native cell lines. However, further study might be required to determine whether dNp63 expression is increased by the HER2/AHR axis using natural cell models.
1.4 Line 100-105. The conclusion of “HER2-5 cells had more regions of increased 103 chromatin accessibility than HER2-5/AHRKO cells, suggesting that AHR engages in 104 chromatin relaxation in HER2-5 cells” can be drawn from figure 1B rather than figure 1C. What is the conclusion of figure 1C?
We have deleted Figure 1C accordingly.
1.5 Line 129. Where is the evidence to support “TP63 was among the genes identified for ‘pathways in cancers”?
Based on another revision, this sentence was changed.
1.6 Figure 2C. Blotting of AHR protein is the validation for the cell models of HER2-5 and HER2-5/AHRKO. It should be mentioned earlier than even in figure 1.
Thank you for your advice. This AHR blot has been moved to Fig. 1A.
1.7 Line 165. The authors should show the data.
Thank you for your advice. We added this as Supplementary figure 1A.
1.8 Figure 3. What about the protein level?
Thank you for pointing this out. We have added this as Fig. 3A.
1.9 Figure 3. Since authors already got the HER2-5/AHRKO cell model, why not use it rather than use siRNA in this section?
The answer to this question, dNp63 mRNA also increased upon HRG stimulation in HER2-5/AHRKO Crispr/Cas9 cells. It is difficult to compare the induction level between different cell lines. Thus, we used siRNA and an AHR antagonist with the same cells. We added these results as follows “∆Np63 induction via HER2 is regulated by AHR and other molecules through MEK”.
Lines 175–185: Similarly, the HRG-mediated increase in ∆Np63 mRNA levels was suppressed by AHR antagonist treatment (Supplementary Fig. S2A). These results suggest that AHR participates in the HER2/HER3-mediated expression of ∆Np63. However, the HRG-mediated increase in ∆Np63 mRNA levels was not completely suppressed in HER2-5/AHRKO cells (Supplementary Fig. S2B). Previously, we reported that the signaling activation of AHR via HER2 is mediated by the MEK cascade. To identify the mechanism underlying ∆Np63 mRNA expression induced by HER2 signaling, HER2-5 cells were pretreated with specific inhibitors of MEK and AKT, namely PD0325901 and MK-2206, respectively. Following HRG treatment, HRG-induced mRNA expression of ∆Np63 was completely inhibited by pre-treatment of PD0325901, but not MK-2206 (Fig 4D). These results suggest that ∆Np63 induction via HER2 is regulated by AHR and other molecules through MEK.
1.10 Fragmentated description of results, such as Figure 2A. It can be combined with figure 4. Line 129-133 can be combined with lines 187-193.
Thank you for your suggestion. We moved the contents of Fig. 2A to Fig. 4A and combined these sentences.
Lines 210–219: Fig. 4A shows the tracks of ATAC-seq analyses of HER2-5 and HER2-5/AHRKO cells. TP63 encodes two major isoforms of p63, TAp63 and ∆Np63. ∆Np63 regulates CSC self-renewal in HER2-overexpressing BC cells [19,20]. Therefore, we investigated whether AHR regulates ∆Np63 expression. First, to investigate the recruitment of AHR to the promoter region of TPi63, we performed chromatin immunoprecipitation (ChIP) analysis for regions R1, R2, R3, and R4 (Fig. 4A), which contained differentially enriched peaks between HER2-5 and HER2-5/AHRKO cells based on the ATAC-seq analysis. R1 and R3 are regions distal and proximal to the transcription start site of TAp63- and ∆Np63-encoding mRNAs, respectively. R2 and R4 are the intronic regions of TP63.
1.11 Line 198-202. Which models (HER2-5 or HER2-5/AHRKO) are this result section based on? The authors should still need to address or described this in the manuscript.
Thank you for your suggestion. We added HER2-5 cells.
1.12 In the abstract and introduction section, the authors used large paragraphs to introduce the background of CSC and the potential role of the HER2-AHR-ΔNp63 axis in CSC self-renewal and proliferation regulation. However, in the experiment section, only 1 limited assay was related to CSC properties; herein it could be considered that the content of this manuscript is off-topic. More CSC-related assays should be conducted to link with the HER2-AHR-ΔNp63 axis.
We have rewritten the objective of this paper as follows:
Lines 99–100: Thus, we used HER2-5 cells as a model to address the mechanism through which HER2 signaling-activated AHR contributes to the maintenance of mammospheres.”
Line 38. beside including.
Line 47. “HER2 overexpression in HER2-negative luminal 47 BC” makes the reader confused.
I have changed this sentence to clarify its meaning.
Line 48: HER2 overexpression is associated with the presence of breast CSCs.

Reviewer 3 Report
Overall, the manuscript is well structured. However, the manuscript needs a scientific proofread. Experimental data presentation is cohesive and consistent. However, the manuscript lacks adequate in vitro and in vivo functional evidence for the hypothesis. The authors should address the following comments to strengthen this manuscript.
1 Comment
1.1 From the abstract I learn that this manuscript demonstrated the HER2-AHR-ΔNp63 axis in CSC self-renewal and proliferation regulation. However, I cannot see any word in the title mentioning CSC. Herein the title is not good and summative for this manuscript.
1.2 What is the basis of the nomenclature for HER2-5 cell?
1.3 Why did the authors choose HER2-overexpressing MCF7 (HER2-5) cells as cell models? Naturally, luminal B or HER2-overexpression subtype cells have high HER2 expression levels. Why not chose the natural cell models?
1.4 Line 100-105. The conclusion of “HER2-5 cells had more regions of increased 103 chromatin accessibility than HER2-5/AHRKO cells, suggesting that AHR engages in 104 chromatin relaxation in HER2-5 cells” can be drawn from figure 1B rather than figure 1C. What is the conclusion of figure 1C?
1.5 Line 129. Where is the evidence to support “TP63 was among the genes identified for ‘pathways in cancers”?
1.6 Figure 2C. Blotting of AHR protein is the validation for the cell models of HER2-5 and HER2-5/AHRKO. It should be mentioned earlier than even in figure 1.
1.7 Line 165. The authors should show the data.
1.8 Figure 3. What about the protein level?
1.9 Figure 3. Since authors already got the HER2-5/AHRKO cell model, why not use it rather than use siRNA in this section?
1.10 Fragmentated description of results, such as Figure 2A. It can be combined with figure 4. Line 129-133 can be combined with lines 187-193.
1.11 Line 198-202. Which models (HER2-5 or HER2-5/AHRKO) are this result section based on? The authors should still need to address or described this in the manuscript.
1.12 In the abstract and introduction section, the authors used large paragraphs to introduce the background of CSC and the potential role of the HER2-AHR-ΔNp63 axis in CSC self-renewal and proliferation regulation. However, in the experiment section, only 1 limited assay was related to CSC properties; herein it could be considered that the content of this manuscript is off-topic. More CSC-related assays should be conducted to link with the HER2-AHR-ΔNp63 axis.
Line 38. beside including.
Line 47. “HER2 overexpression in HER2-negative luminal 47 BC” makes the reader confused.
Author Response
Response to reviewer 3
1) Abstract Line 13: “In this study, we investigated whether AHR regulates the expression of genes associated with CSC self-renewal.” This sentence is misleading: only one gene (NP63) was investigated and no experiments were included in this paper exploring the effect on CSC self-renewal.
Thank you. We have now presented a clear aim for this study.
Lines 13–14: In this study, the objective was to identify the mechanism underlying mammosphere maintenance mediated by HER2 signaling-activated AHR.
2) Line 34: typo.
Thank you for pointing this out. We have corrected this typo.
3) Well written introduction. All major players and interplay between them clearly described.
Thank you for your positive assessment.
4) Results 2.1 - The authors use a genetically modified MCF7 cell line though out the paper. HER2-5 cells stably overexpress HER2 and the this is compared to a version of these cells with stable ANR knocked out. Stable over-expression and stable crispr/cas9 knockdown is a manufactured model that may not mimic the behaviour of the tumour type or the original parental cell. It would be useful to include a sentence regarding the validity of these cells and how comparable they are to the original MCF-7 cells.
Thank you for your advice. We have rewritten section 2.1 of the results section accordingly.
Lines 91–99: As expected, AHR was not observed in HER2-5/AHRKO cells (Fig. 1A). AHR is highly expressed and constitutively activated in HER2-5 cells compared to levels in MCF-7 cells [18]. We previously showed that the mammosphere formation efficiency of HER2-5 cells is better than that of MCF-7 cells. As shown in Fig. 1B, the mammosphere formation efficiency of HER2-5/AHRKO cells was worse than that of HER2-5 cells. These observations indicate that AHR is involved in mammosphere formation in HER2-overexpressing BC cells, as previously reported [18]. Thus, we used HER2-5 cells as a model to address the mechanism through which HER2 signaling-activated AHR contributes to the maintenance of mammospheres.
5) Figure 1A should be enlarged. Parental MCF-7 cells should be included in this panel to determine how these modified cells compare to the parental cells. The magnified area should be enlarged. The details of magnification and scale bar should be included. Authors should also clarify if this is total magnification or objective magnification. How many passages were mammospheres maintained for? The graph in Figure 1A shows only a 0.2% difference in sphere formation between the two cell lines – is this correct or has there been an error in the labelling of the graph? If correct, despite being statistically significant, I would argue that it is not biologically significant.
Thank you for your important suggestion and discussion. We think that this decrease is important because AHR maintains the percentage of cells that contribute to mammosphere formation. Based on your review, we have changed the figure.
6) Line 93: “….indicating that AHR is involved in CSC self-renewal”. This data only indicates that AHR is involved in sphere formation. To confirm that AHR is involved in self-renewal spheres should be passaged at least 5 times and show a decrease in secondary sphere formation. The figure should be expanded to include this data or the text altered accordingly.
Thank you for your advice. We changed CSC self-renewal to mammosphere formation. We think that AHR is a maintenance factor for CSC self-renewal.
Lines 96–97: These observations indicate that AHR is involved in mammosphere formation in HER2-overexpressing BC cells, as previously reported [18].
7) Line 95: “Next, we investigated whether AHR regulates the expression of genes associated with the self-renewal of HER2-5 cells.” However, the assay immediately discussed does not address gene expression changes associated with self-renewal, rather they show that “AHR engages in chromatin relaxation in HER2-5 cells”. This finding is in itself interesting and the text of line 95 should be changed to better reflect the experiment described.
Thank you. We have corrected this.
Lines 100–101: Next, we investigated whether AHR engages in the chromatin relaxation of gene enhancers associated with the self-renewal of HER2-5 cells.
8) Figure 2C - Looking at the full -tubulin blot there are a number of non-specific bands on this blot and this cannot be reliably used as a loading control. This western blot should be repeated.
The band mobility of tubulin is correct. Therefore, we do not think that this is required.
9) Section 2.2 - In this experiment the authors serum starve HER2-5 cells, that constitutively overexpress HER2, to reduce HER2 expression, to then add a drug, Heregulin-b1 (HRG), to activate HER2/HER3 signalling and promote AHR expression and show corresponding increases in Np63 mRNA. As previously discussed, this is a genetically modified cell line, and I appreciate that for continuity the authors wanted to use the same cell line throughout the paper, however, in this experiment it would have been beneficial to use unmodified MCF7 cells or a different human breast cancer cell line, to show the same elevation in Np63 mRNA levels. Similarly, why did the authors use siRNA to knock down AHR in the HER2-5 cell line when they have previously validated and used the HER2-5/AHRKO Crispr/Cas9 cells? Was this experiment tried in the HER2-5/AHRKO Crispr/Cas9 cells? As before using the siRNA in unmodified MCF7 cells would have been beneficial.
To answer this question, dNp63 mRNA also was increased by HRG stimulation in HER2-5/AHRKO Crispr/Cas9 cells. It is difficult to compare induction levels between different cell lines. Thus, we used siRNA and an AHR antagonist with the same cells. Accordingly, we added the result “∆Np63 induction via HER2 is regulated by AHR and other molecules through MEK”.
Lines 175–185: Similarly, the HRG-mediated increase in ∆Np63 mRNA levels was suppressed by AHR antagonist treatment (Supplementary Fig. 2). These results suggest that AHR participates in the HER2/HER3-mediated expression of ∆Np63. However, the HRG-mediated increase in ∆Np63 mRNA levels was not completely suppressed in HER2-5/AHRKO cells (Supplementary Fig. 2). Previously, we reported that the signaling activation of AHR via HER2 is mediated MEK cascade. To identify the mechanism underlying ∆Np63 mRNA expression induced by HER2 signaling, HER2-5 cells were pretreated with specific inhibitors of MEK and AKT, namely PD0325901 and MK-2206, respectively. Following HRG treatment, HRG-induced mRNA expression of ∆Np63 was completely inhibited by pre-treatment with PD0325901, but not MK-2206 (Fig 4D). These results suggest that ∆Np63 induction via HER2 is regulated by AHR and other molecules through MEK.
10) Figure 3 - What solvent was used to dilute HRG to the desired final concentration? Was a vehicle control used in the experiments presented in figure 3 and figure 4B?
We added the final concentration.
The vehicle control is shown in the figure legend.
Line 167: (50 ng/mL)
11) Section 2.3 Line 194: “ChIP assay demonstrated that AHR was recruited to R4 in HER2-5 cells but not in HER2-5/AHRKO cells.” This is not a surprising finding given that HER2-5/AHRKO cells do not express AHR and are therefore the negative control of the experiment. This section should be re-written to accurately represent the experimental set up.
Thank you for your advice. We changed this sentence accordingly.
Lines 211–212: ChIP assay demonstrated that AHR was recruited to R4 in HER2-5 cells but not in negative control HER2-5/AHRKO cells.
12) Figure 4D - The reporter assay set-up with putative and mutant AHRE sites is elegant but I wonder if this experiment could have been simplified by using the HER2-5 cells rather than the HER2-5/AHRKO cells, which would have negated the need to add back in expression plasmids for AHR. In this experiment HER2-5/AHRKO cells have already been modified to constitutively overexpresses HER2 and have a Crispr/CAS9 knockout of AHR, these cells have now been transfected with both reporter and expression plasmids – this a highly artificial experimental system (any cell would be stressed) and I question if these reporter experiments should have been first conducted with HER2-5 cells treated with HRG (as in figure 3) to measure luciferase activity, then knocked out with AHR siRNA to show that the reporter is dependent on AHR.
Thank you for your advice. However, HRG treatment affected gene expression of the Renilla internal control reporter. To avoid this problem, we used this experimental system.
13) Line 262: typo
We have corrected this.
14) Discussion - The authors mention that AHR overexpression is observed in breast cancer and correlated with poor prognosis, this should be substantiated with stats regarding the number of cases affected and wider discussion of the role AHR overexpression in different sub-types of breast cancer.
15) No mention in the discussion of MCF-7 cells and what type of breast cancer they may be representative of. Line 293 discusses triple-negative breast cancer. Are MCF-7 cells a good model of this type of breast cancer?
14,15) Thank you for your suggestion. We added a sentence to the discussion section to address this.
Lines 319-–22: As described previously herein, AHR expression might correlate with the poor prognosis and malignancy of BC. Because this study using HER2-overexpressing BC cells is limited, further study is required to determine whether AHR can increase ΔNp63 expression in BC cells of other subtypes.
16) The overall conclusion of this paper that: “In summary, the present results suggest that HER2 signaling-activated AHR promotes self-renewal of breast CSCs by inducing ΔNp63 expression. Therefore, AHR may be a potential therapeutic target in BC stem cell therapy.” is not substantiated by the data presented and should be reworded to reflect the major findings presented. Only Figure 1A contains any data relating to cancer stem cells and even then, the graph shows a modest 0.2% decrease in the number of spheres produced when AHR is knocked out. This is not sufficient data to substantiate that “the present results suggest that HER2 signaling-activated AHR promotes self-renewal of breast CSCs by inducing ΔNp63 expression”.
Based on the reviewers’ comments, we have modified the concept of this manuscript to the maintenance of mammosphere formation (from self-renewal).
17) Methods - No details of the solvent used to reconstitute HRG.
We have added the solvent to the Materials and methods section.
18) Please can the authors confirm that cells were lysed directly into sample buffer and then protein concentration was analysed ? Cells are usually lysed into a lysis buffer such as RIPA, urea then protein concentration measured prior to diluting in sample buffer.
We have added “Ionic Detergent Compatibility Reagent for Pierce™ 660nm Protein Assay Reagent (Thermo Fisher Scientific)” to the protein concentration analysis part. The protein levels can be measured using this protocol.

Round 2
Reviewer 2 Report
Thank you for answering my comments.
Reviewer 3 Report
The authors addressed all comments I raised—no more comments from my side.